# Intraspecific Genetic Variation for Behavioral Isolation Loci in *Drosophila*

**DOI:** 10.3390/genes12111703

**Published:** 2021-10-26

**Authors:** Jessica A. Pardy, Samia Lahib, Mohamed A. F. Noor, Amanda J. Moehring

**Affiliations:** 1Department of Biology, The University of Western Ontario, London, ON N6A 5B7, Canada; jpardy4@uwo.ca (J.A.P.); slahib@uwo.ca (S.L.); 2Biology Department, Duke University, Durham, NC 27708, Canada; noor@duke.edu

**Keywords:** behavior, genetic variation, quantitative genetics, reproductive isolation, speciation

## Abstract

Behavioral isolation is considered to be the primary mode of species isolation, and the lack of identification of individual genes for behavioral isolation has hindered our ability to address fundamental questions about the process of speciation. One of the major questions that remains about behavioral isolation is whether the genetic basis of isolation between species also varies within a species. Indeed, the extent to which genes for isolation may vary across a population is rarely explored. Here, we bypass the problem of individual gene identification by addressing this question using a quantitative genetic comparison. Using strains from eight different populations of *Drosophila simulans*, we genetically mapped the genomic regions contributing to behavioral isolation from their closely related sibling species, *Drosophila mauritiana*. We found extensive variation in the size of contribution of different genomic regions to behavioral isolation among the different strains, in the location of regions contributing to isolation, and in the ability to redetect loci when retesting the same strain.

## 1. Introduction

Species are isolated from one another by reproductive barriers that limit gene flow, allowing each species to evolve along a separate trajectory. One of the first reproductive barriers to arise between species is behavioral isolation [1,2], which prevents mating between males and females from different species. Behavioral isolation is thought to have a greater effect on reproductive isolation as it acts earlier in the reproductive cycle [3] and is more often found to be the only isolating mechanism between species [1,4]. This barrier arises due to divergence in the two species for traits necessary for successful recognition and acquisition of mates, such as differences in pheromone composition (e.g., [5,6]) or courtship song (e.g., [7,8]). While the results of speciation are usually easily observed, and the ecological basis of many speciation events have been well characterized, the genetic mechanisms via which it occurs remain unclear.

The well-characterized courtship behavior of *Drosophila*, paired with the wealth of genetic information available for many *Drosophila* species, makes this system an ideal and appropriate model for behavioral isolation studies. The traits that may underlie species isolation among *Drosophila* often vary within species. For example, the composition of cuticular hydrocarbons (CHCs), used both as pheromones and for desiccation resistance in *Drosophila*, can vary clinally among populations [9,10] and diverge between species [11,12]. Similarly, courtship song has been shown to vary within (e.g., [13]) and between (e.g., [14]) species. Within-species variation can influence the potential for incipient speciation between populations [15,16], as well as the strength of isolation between different populations of a species with a sister species [17,18]. While the genetic basis of behavioral variation within a species could reasonably be assumed to also contribute to behavioral isolation between species, there is some evidence that the underlying basis of these two traits is instead genetically distinct [19,20,21], but more studies are needed.

The quantitative genetic basis of behavioral isolation between *Drosophila* species has been mapped in a number of studies (for reviews, see [3,22]). While these studies have not been refined to the level of individual genetic loci, comparisons of the mapped regions have allowed for an assessment of broad evolutionary questions. For example, the genomic regions contributing to behavioral isolation were largely mapped to regions of low recombination [23]. While the loci for male traits and female preference do not generally appear to co-localize in the genome [15,24,25,26,27,28,29], these maps are often not refined enough to exclude the possibility of colocalization, and there is evidence that it can occur for the traits and preferences underlying behavioral isolation [30].

One of the most widely studied genetic model systems for studying reproductive isolation in *Drosophila* is the species pair *D. simulans* and *D. mauritiana*, which are members of the melanogaster subgroup. *D. simulans* is generally found worldwide and is allopatric to *D. mauritiana*, which is endemic to the island of Mauritius in the Indian Ocean [25]. The genomes of these two species are almost entirely homosequential, and individual flies can only be phenotypically distinguished from one another morphologically by the shape of the male genital arch [25]. When paired, these two species exhibit asymmetrical sexual isolation [25]: *D. mauritiana* females rarely mate with *D. simulans* males, while *D. simulans* females will readily mate with *D. mauritiana* males. While this mate discrimination has been well documented, the individual genetic bases of the *D. mauritiana* female discrimination and the *D. simulans* male traits they are discriminating against remain largely unknown.

A previous study that mapped the genetic basis of behavioral isolation between *D. simulans* and *D. mauritiana* found seven loci in females for *D. mauritiana* female preference, found on all three major chromosomes, and three loci in males for the *D. simulans* male trait being discriminated against, all on the third chromosome [31]. Using the same strains of the two species, a later study confirmed two of these regions using introgressions (the third was not tested) and found a genetic linkage between the male trait and female preference loci [30]. Here, we expand upon these previous findings to examine whether the loci contributing to behavioral isolation between two species also vary within a species, and whether the genetic contributions to behavioral isolation are robust to subtle variations in a behavior assay. We focus solely on the *D. simulans* male traits these females are discriminating against, since this species has a worldwide distribution with a greater potential for population variation, while *D. mauritiana* is endemic to a small island. We focus on the third chromosome since all of the regions exhibiting significant associations with the male behavioral isolation traits were located on this chromosome in the original quantitative trait locus (QTL) map.

We first assessed whether there was variation in the strength of behavioral isolation between *D. mauritiana* females and males from different strains of *D. simulans*. We next performed QTL mapping for courtship and copulation traits in strains from three populations of *D. simulans*: the strain used in the original mapping study (*simulans* FC: [31]), a strain conferring relatively high mating of F_1_ hybrid males with *D. mauritiana* females, and a strain conferring relatively low mating of F_1_ hybrid males with *D. mauritiana* females. We then repeated these experiments using a slightly different assay paradigm on the same three strains of *D. simulans* used above, plus five additional strains of *D. simulans*. A comparison of the genetic regions (QTL) contributing significantly to behavioral isolation in these different conditions and across different strains allows for the assessment of whether the same loci or different loci contribute to intraspecific variation in interspecific mating success, and how robust the genetic maps of this trait are when there is a subtle variation in protocol.

## 2. Materials and Methods

### 2.1. Drosophila Stocks and Crosses

All stocks, crosses, and collected virgins were maintained on a 14:10 h light–dark cycle at 24 °C and 75% relative humidity in 30 mL (8 dram) vials containing approximately 7 mL of standard Bloomington recipe fly food (Bloomington *Drosophila* Stock Center). Six lines of *D. simulans* were obtained from the *Drosophila* Species Stock Center: *sim*194 (14021-0251.194 from Winters, California), *sim*196 (14021-0251.196 from Ansirable, Madagascar), *sim*197 (14021-0251.197 from Joffreville, Madagascar), *sim*198 (14021-0251.198 from Noumea, New Caledonia), *sim*199 (14021-0251.199 from Nanyuki, Kenya), and *sim*216 (14021-0251.216 from Winters, California). *D. simulans* Florida City (*sim*FC from Florida City, Florida; [25]), *D. simulans* 167.4 (*sim*167.4 from Nanyuki, Kenya), and *D. mauritiana* Synthetic (*mau*SYN; [25]) were provided by J. Coyne. The *D. simulans* strains were selected to represent both a range and a replication of geographic locations.

For all crosses, 6–10 virgin females were paired with 6–10 virgin males, all aged 5–7 days. We first assessed whether there was variation among strains in the degree to which *D. simulans* males are rejected by *D. mauritiana* females. Since *D. mauritiana* females reject pure-species *D. simulans* males 100% of the time in our assays, we assayed the strain variation using F_1_ interspecies males. Virgin *D. simulans* females of the eight strains were each crossed to *mau*SYN males, producing F_1_ males for assays.

Male F_1_ hybrids between these two species are sterile, but female F_1_ hybrids are fertile [25] and can be used to create recombinant backcross offspring. In the first round of backcross experiments, virgin *D. simulans* females of *sim*197, *sim*216, and *sim*FC were each crossed to *mau*SYN males. These lines were chosen because they had the lowest tendency to mate with *D. mauritiana*, had the highest tendency to mate with *D. mauritiana*, and were the original line tested [31], respectively. The resulting F_1_ females were collected as virgins and then backcrossed (BC) to *mau*SYN males, creating three populations of males used in behavioral assays and subsequent QTL mapping: BC_197_, BC_216_, and BC_FC_, respectively.

In the second round of backcross experiments, conducted 2 years later, virgin *D. simulans* females of each of the eight lines were crossed to *mau*SYN males; the resulting F_1_ females were backcrossed to *mau*SYN males to make eight populations of backcross males used in behavior assays and mapping: BC_167,_ BC_194_, BC_196_, BC_197_, BC_198_, BC_199_, BC_216_, and BC_FC_. For all backcrosses, at least 15 independent vials of cross were made for each backcross type. The BC generation possesses one homolog of each chromosome that is entirely *D. mauritiana*, and the other homolog of each chromosome is a unique mix of *D. mauritiana* and *D. simulans* genetic material created through recombination in the F_1_ female.

### 2.2. Behavior Assays

No-choice behavioral assays were performed to determine whether male flies were successful in copulating with *D. mauritiana* female flies. All tested flies were collected as virgins and aged 5–7 days prior to behavioral assays to ensure reproductive maturity. All assays occurred at room temperature (22–24 °C) between 2 and 3 h of ‘lights on’.

For the initial pure-species and F_1_ assays, a single male and a single *mau*SYN female were aspirated into an 8 dram polystyrene vial containing approximately 2 mL of food medium. Flies were assayed for 45 min in groupings of ~20–30 being observed simultaneously, with equal or near-equal representation from each strain in each assay to control for environmental effects (e.g., [32]). Whether courtship and copulation occurred was scored. We scored 20 pure-species and 39 F_1_ males from each strain.

In the first round of BC male assays, a single male and a single *mau*SYN female were aspirated into an 8 dram polystyrene vial containing approximately 2 mL of food medium. Flies were assayed for 1 h in groupings of ~40–60, with equal or near-equal representation from each strain. We scored 493 BC_197_, 491 BC_216_, and 507 BC_FC_ males. The original assay using BC_FC_ [31] only found significant QTL for the proportion of males that copulated out of those that courted; thus, we used that same metric here of “copulation occurrence” being analyzed only for those assays in which the male courted the female (*N* = 292 BC_197_, 349 BC_216_, and 332 BC_FC_). Only those that courted (then mated or did not mate) were used for this metric to ensure that the assay was scoring *mau* female choosiness against courting males, and not scoring whether or not a male chose to court a female. To test whether other aspects of mating behavior are also consistent across strains, we additionally scored the time until first courtship (courtship latency), time until successful copulation (copulation latency), and the time from the start to the end of copulation (copulation duration). Note that these traits did not have any significant QTL in the original study’s assay of BC_FC_ males. After the assay, BC males were promptly frozen and stored at −20 °C.

We performed the second round of assays under slightly different environmental conditions to assess whether QTL are robust across subtle environmental variations. In the second round of assays, a single BC male and a single *mau*SYN female were aspirated into an 8 dram glass vial sprayed with a light mist of water to provide humidity. Flies were assayed for 1 h in groupings of ~60–100, with equal or near-equal representation from each strain in each assay to control for environmental effects. Males that courted were scored for copulation occurrence; males that did not court were discarded and not genotyped. Approximately 150 individual males that courted were scored for each of the eight *D. simulans* lines (BC_167_ = 147_,_ BC_194_ = 151, BC_196_ = 155, BC_197_ = 150, BC_198_ = 156, BC_199_ = 160, BC_216_ = 152, and BC_FC_ = 152), for a total of 1223 BC male flies. As above, BC males were promptly frozen and stored at −20 °C for later genotyping.

### 2.3. Molecular Markers

DNA was extracted from the assayed BC males using Engel’s method [33], standard PCR was used to amplify a single marker or multiplex PCR was used to amplify several markers simultaneously [34], and then the products were visualized using either a 3% agarose gel or fragment analysis, respectively (see below). A touchdown PCR protocol was followed [35], with annealing temperatures as follows: 95 °C 5 min, three cycles 94 °C 1 min/55 °C 30 s/72 °C 30 s, three cycles 94 °C 1 min/53 °C 30 s/72 °C 30 s, 30 cycles 94 °C 1 min/50 °C 30 s/72 °C 30 s, 5 °C 3 min.

The first round of assays was genotyped for nine molecular markers on the third chromosome (Appendix A). These microsatellite markers had a size difference between the two species; thus, BC individuals would either be homozygous for the *D. mauritiana* genome and have one product on an agarose gel or would be heterozygous and have two differently sized products. To confirm expected band sizes, DNA samples extracted from each of *D. simulans, D. mauritiana*, and an F1 *simulans/mauritiana* hybrid were used as controls. Some microsatellites did not have a notable size difference between a particular strain of *D. simulans* and the size in *D. mauritiana*; an alternate nearby microsatellite was used in these cases. The cytological locations of markers (based on *D. melanogaster* cytology) were as follows: 62B, 63E, 73C, 78D, 82D, 84D/E, 90B, 97D, and 100E. The markers were intentionally clustered around the centromere and telomeres as this is where the initial genetic map identified significant regions contributing to behavioral isolation.

The second round of assays was also genotyped, using 12 microsatellite markers (Appendix A), but the size differences were assessed using fragment analysis rather than agarose gels, as this method allowed for amplifying all markers simultaneously. In fragment analysis, each of the primer pairs is fluorescently labeled with a different fluorophore, allowing amplified products to be distinguished when run together. Following PCR, samples were dehydrated using an Eppendorf Vacufuge and then shipped to the NAPS Unit at UBC for analysis. Genemapper software was used to classify each individual as either homozygous or heterozygous. The same controls were used as above. Any individual marker genotypes that did not amplify via fragment analysis were reamplified individually via PCR and visualized on an agarose gel. The cytological locations of markers were as follows: 61B, 62B, 63E, 73C, 78D, 82D, 84E, 93E, 94A, 95F, 97F, and 100E.

### 2.4. QTL Mapping

All raw behavior and genotyping data can be found in Appendix A. The raw data from [31] can be found in Appendix A. QTL were mapped with this data using composite interval mapping (CIM; [36]) as implemented by QTL Cartographer software [37]. This software calculates the likelihood ratio (LR), which is the likelihood of a gene for the trait of interest being present in the interval between two markers. LR values were calculated every 2 cM with marker cofactors at distances greater than 10 cM. The threshold above which the LR value is statistically significant at *p* ≤ 0.05 were calculated separately for each line and trait using 1000 permutations of the data, which takes marker correlations and multiple testing effects into consideration [38,39]. Since we wanted a permissive, rather than restrictive, comparison, we also calculated significant regions with the less stringent cutoff of *p* ≤ 0.1 for the strains that were tested, since their sample size was smaller than that of the original study. While the score of copulation occurrence violates the assumption of normality for CIM mapping since it is a binary measure, CIM has previously been shown to be very robust to departures from normality (e.g., [31]. QTL boundaries were estimated from the threshold determined by permutations or the 2-LOD support intervals (*p* < 0.05), whichever was reached first. QTL effects were calculated at each LR peak as the difference between the phenotypes of heterozygotes vs. homozygotes, scaled by the standard deviation. The proportion of the variance accounted for by the QTL was calculated as *R*^2^ = (*s*_0_^2^ − *s*_1_^2^)/*s*^2^, where *s*_0_^2^ is the sample variance of the residuals, *s*_1_^2^ is the variance of the residuals, and *s*^2^ is the variance of the trait; this value is then scaled by the cofactors in the model to give the adjusted proportion of the variance accounted for by the QTL (called *TR*^2^).

For the first round of assays, using three strains of *D. simulans*, we tested for epistasis between loci for mating success in males using multiple interval mapping (MIM) in QTL Cartographer. We did not perform this analysis for the second round of assays on eight strains as the sample size was too small. We used a window size of 10, walk speed of 1 cM, and a significance threshold of 0.10. It was previously shown that a threshold of 0.10 does not compromise the number of false positives and is more likely to capture significant effects than a threshold of 0.05 [40]. We attempted to identify epistasis between QTLs identified using CIM, as well as between QTLs and genomic regions that did not contain a significant QTL.

We compared whether the same three significant QTL in the original map of BC_FC-ORIG_ also impacted copulation occurrence when compared to the first or second rounds of assays with BC_FC_ in the current study. We used a three-way contingency test based on a log-linear model [41] with the groups being strain (BC_FC-ORIG_ vs. BC_FC_), genotype (heterozygous vs. homozygous), and copulation occurrence (yes vs. no). We used the raw data for the markers closest to the three QTL peaks in BC_FC-ORIG_ [31] and compared to the data for the closest markers to those same BC_FC-ORIG_ peak locations in the new assays of BC_FC_. We repeated this same analysis across the eight strains used in round 2.

## 3. Results

We first quantified whether there was variation in behavioral isolation between different strains of *D. simulans* and *D. mauritiana*. Although qualitatively almost all of the *D. simulans* males of the eight strains we tested courted a *D. mauritiana* female, none of them copulated within our 1 h assay (*N* = 20), indicating that mating rates were necessarily quite low among all strains. We, therefore, tested F_1_ interspecies males made from each of the eight strains to determine if there was genetic variation across the strains for mating success with *D. mauritiana* females. We found that 77–95% of F_1_ males courted *D. mauritiana* females, and, of those males that courted, 10–57% achieved copulation (Table 1).

In the first round of assays, we tested backcross (BC) males made from the strain used in the original study (BC_FC_; [31]), and the strains we found had the lowest (BC_197_) and highest (BC_216_) amount of mating with *D. mauritiana* (Table 1). We scored the BC males made from these three lines for their mating success with *D. mauritiana* females when assayed in the presence of food. In the second round of assays, we again tested BC males from the above three strains, plus the remaining five strains listed in Table 1, for their mating success with *D. mauritiana* females, but this time assayed in the presence of moisture but no food. We then genotyped and performed QTL mapping to identify genomic regions affecting mating success.

Copulation occurrence, which mapped strongly in the original study (BC_FC-ORIG_; Appendix A), was not significant for any regions when we retested backcross males made from the same line using either the same (round 1) or different (round 2) assay conditions (Table 2; Figure 1; Appendix A). The results were also significantly (*p* < 0.0001 for all) different between BC_FC-ORIG_ and the first or second round of assays here of BC_FC_ for each of the three original QTL peaks when compared using a three-way contingency table (66C: G^2^ = 51.2, G^2^ = 35.1; 82C: G^2^ = 86.5, G^2^ = 104.5; 97B: G^2^ = 58.4, G^2^ = 49.3; first and second assays, respectively). While the other two strains that were tested under two conditions (BC_197_ and BC_216_) had significant regions for copulation occurrence, these regions varied by assay condition. When looking across all eight strains under the same assay condition (e.g., across condition #2), there was very little overlap, with no region shared among three or more strains (Table 2; Figure 1; Appendix A). Similar to the above comparison of one strain across the three assay conditions, here, we compared the QTL peaks across the eight strains used in round 2 assays. We chose the molecular marker closest to the significant peaks found in any strain for copulation occurrence (62B, 67E, 84E, 93E). As above, the results were significantly (*p* < 0.0001 for all) different across the eight strains when compared using a three-way contingency table.

To test whether additional traits other than copulation success are consistent across strains, we also scored three strains (BC_197_, BC_216_, and BC_FC_) for the additional mating behaviors of courtship latency, copulation latency, and copulation duration. Note that none of these behavioral traits mapped a significant region in the original assay of BC_FC-ORIG_. Two lines (BC_FC_ and BC_216_) also mapped loci for copulation latency but the regions did not overlap (Table 3; Appendix A). Copulation duration mapped loci in all three of the populations tested here, with the significant regions overlapping. Two of the strains (BC_197_ and BC_216_) had significant regions on the third chromosome for courtship latency, which was not measured in the original study, and this region overlapped. Similarly to what was found in the original study [31], none of the traits showed epistasis between QTL or between the QTL and nonsignificant regions for males from any of the lines.

## 4. Discussion

The division of populations into defined species is thought to occur when natural selection acts on variation in a trait affecting gene flow among these different populations. These incipient species then become reproductively isolated from one another and diverge into two distinct species. Mating behavior varies within *D. simulans* when males are paired with conspecific females [42]; therefore, we thought it likely that there would also be underlying genetic variation in the male mating behaviors that affect species isolation. By comparing the regions in *D. simulans* males that *D. mauritiana* females discriminate against in eight different lines of *D. simulans*, we determined whether the underlying genetic basis of species isolation is consistent or varies across a species.

We predicted that there would be some genomic regions that were consistent across strains, due to common ancestry from the original speciation event. Likewise, we expected that variation in some loci would have arisen across strains due to variation that arose after the species expanded to a worldwide distribution. Since *D. mauritiana* females strongly and consistently reject *D. simulans* males [25], we also predicted that these loci would be robust across subtly different environmental conditions. Lastly, we predicted that loci affecting behavioral isolation would be found in regions of low recombination, i.e., near the centromere and telomeres. Our predictions were not confirmed. Only one of the nine identified regions was at a centromere or telomere. We found that different strains tested across the same conditions had no consistent QTL among them and, indeed, had almost no overlap at all in genomic regions contributing to male interspecies copulation success. Likewise, we found that the same strains tested using slightly different conditions yielded different QTL for male interspecies copulation success. It is possible that the strength of effect of loci may vary across different populations and, therefore, that greater overlap would be found with larger sample sizes that could detect loci of smaller effect. If so, then the conclusion might not be that the loci differ among populations, but rather that their effect sizes differ among populations. This possibility requires further exploration. A second possibility is that the populations of *D. simulans* originally had a shared genetic basis for male traits leading to behavioral isolation, but that loci were gained and lost over evolutionary time, leading to divergent bases today. Lastly, it is possible that different populations did not share male variants that affect behavioral isolation and, instead, independent variants arose over time that affect male traits in ways that contribute to behavioral isolation.

In contrast to copulation occurrence, there was consistency in the genetic basis of copulation duration across populations of *D. simulans*. Pure-species *D. simulans* and *D. mauritiana* have notably different copulation duration (20–26 min vs. 11–17 min, respectively; [43,44,45]). The shared genetic basis of this time difference across the three strains we tested indicates a potentially robust genetic basis of this difference. However, this conclusion has the caveat in that one of the same strains (*sim*FC) did not yield any QTL at all for this trait when it was assessed at an earlier time [31]. It is possible that, although this strain had the same collection origin, its separate maintenance as a stock in different locations in the intervening years yielded enough genetic divergence to produce these varying genetic mapping results. Regardless of the cause, the major loci contributing to mating behavior traits, at least as produced via QTL mapping, do not appear to be robust to subtle changes in strains or assay conditions. This does not negate the usefulness of QTL studies for identifying candidate loci for traits of interest, but does demonstrate the importance of performing follow-up studies for fine mapping and confirming causal genes using the exact same strains and conditions as used in the original study. It also indicates that caution must be exercised in extrapolating results outside of the specific assay conditions and population studied.

These findings add to previous work (e.g., [17,18]) that demonstrate that loci for behavioral isolation between species may have a high amount of variation within species, where different loci are the primary contributors to behavioral isolation in different populations. Since female *D. mauritiana* consistently display rejection toward all strains of *D. simulans* that we tested, our genetic mapping results suggest that the male behavioral traits that isolate these closely related species may occur through different means. They may be caused by the same male traits being generated via different molecular pathways in the different strains or be caused by female discrimination against males using one or more combinations of different characteristics. Regardless of the cause, the presence of variation among strains for isolation between species means that loci that are identified for behavioral isolation using one strain may not contribute to isolation across the entire species.

## 5. Conclusions

We find that loci underlying behavioral isolation have extensive genetic variation within a species and can be influenced by subtle differences in the environment. Therefore, when presenting results of genetic mapping on a single strain in a single environment, caution must be used in using those results to extrapolate the genetic basis of behavioral isolation across a species and across a range of conditions.

## Figures and Tables

**Figure 1 genes-12-01703-f001:**
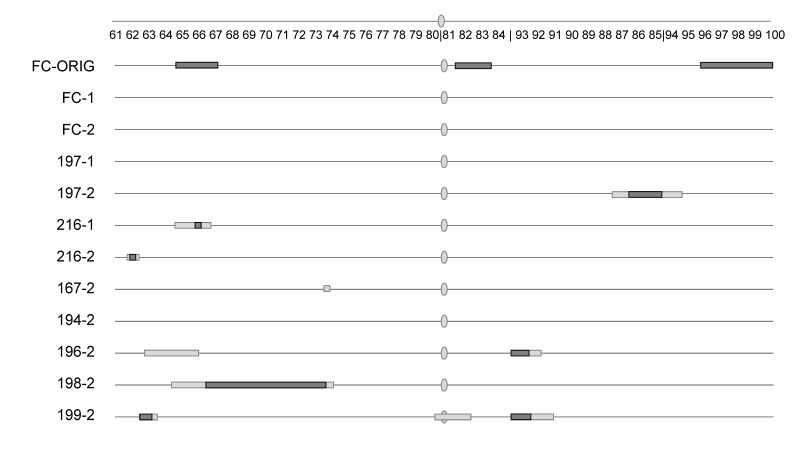
Comparison of significant QTL regions for copulation occurrence across two assay conditions (1 and 2). One line was tested in the original (FC-ORIG) study by [31]. Three lines (FC, 197, and 216) were tested under these same conditions as used in the original study (FC). Eight lines were tested under similar conditions, but without food present. The third chromosome is shown as a line with an oval at the centromere. Cytological region designations based on *D. melanogaster* are shown at the top, with a vertical line at the major landmarks of the centromere and the inversion (from 84F to 93F) breakpoints. Regions that are significant are shown as boxes: threshold of *p* < 0.05 (dark gray) or threshold of *p* < 0.1 (light gray).

**Table 1 genes-12-01703-t001:** Mating frequencies for F1 interspecies hybrid males paired with *D. mauritiana* females.

Male.	*N*	Court	Prop Court	Cop	Prop Cop of Court
*sim*167/*mau*	39	30	0.77	10	0.33
*sim*194/*mau*	39	35	0.90	19	0.54
*sim*196/*mau*	39	37	0.95	11	0.30
*sim*197/*mau*	39	31	0.79	3	0.10
*sim*198/*mau*	39	35	0.90	10	0.29
*sim*199/*mau*	39	37	0.95	14	0.38
*sim*216/*mau*	39	37	0.95	21	0.57
*sim*FC/*mau*	39	34	0.87	12	0.35

**Table 2 genes-12-01703-t002:** QTL for male copulation occurrence between backcross males made from eight strains of *D. simulans* paired with *D. mauritiana* females.

Line	Assay # ^2^	*N*	Region ^3^	Peak ^3,4^	Peak LR ^4^	*TR2* ^5^
BC_FC-ORIG_ ^1^	-	1002	64D–67A	66C	21.19	0.135
			81B–83E	82C	70.31	0.121
			95D–100E	97B	27.06	0.141
BC_FC_	1	332	none	-	-	-
BC_FC_	2	152	none	-	-	-
BC_197_	1	292	none	-	-	-
BC_197_	2	150	84F–86E (84F–88B; 94A–94E)	84F	7.87	0.119
BC_216_	1	349	65E (64C–66E)	65E	7.53	0.065
BC_216_	2	152	61E–62B (61C–62B)	61F	8.83	0.106
BC_167_	2	147	(73C–73D)	73C	6.17	0.049
BC_194_	2	151	none	-	-	-
BC_196_	2	155	(62D–66A)	63E	7.08	0.102
			92F–93E (92B–93E)	93E	9.62	0.104
BC_198_	2	156	66B–73E (64B–74B)	69F	10.54	0.136
BC_199_	2	160	62B–63D (62B–63E)	62E	8.55	0.118
			(80B–82D)	81D	7.57	0.145
			92C–93E (91A–93E)	93E	8.55	0.116

^1^ BC_FC-ORIG_ regions are from [31]. ^2^ Three of the strains were assayed twice (assay #1 and #2). ^3^ The region and peak cytological locations are listed in relation to *D. melanogaster*; note that there is a large inversion from 84F to 93F in relation to *D. melanogaster*. Significant regions are listed for a threshold of 0.05, with a threshold of 0.1 shown in parenthesis for all but BC_FC-ORIG_, which only reported the threshold of 0.05. ^4^ The peak is the cytological location with the highest likelihood ratio (LR). ^5^ *TR2* is the proportion of the variance explained by the QTL when cofactors are taken into account.

**Table 3 genes-12-01703-t003:** QTL for male mating behaviors isolating *D. simulans* from *D. mauritiana*, assessed using backcross males made from three strains of *D. simulans*.

Trait	Line	Region ^1^	Peak ^1,2^	Peak LR ^2^	Effect ^3^	*TR* ^4^
Courtship latency	BC_197_	78D–84E; 93F–91A	93E	10.35	5.65	0.036
	BC_216_	92F–88A	91C	8.51	4.54	0.045
Copulation latency	BC_FC_	84D–84E; 93F–92B	93B	7.60	7.60	0.020
	BC_197_	n/a	n/a	n/a	n/a	n/a
	BC_216_	68F–69A	68F	7.34	−17.47	0.067
		81A–83F	82C	9.81	15.73	0.025
Copulation duration	BC_FC_	telomere-65E	62B	9.94	−2.76	0.110
	BC_197_	telomere-68E	65D	11.54	−3.94	0.179
	BC_216_	telomere-68F	63E	15.74	−3.07	0.096

^1^ The region and peak cytological locations are listed in relation to *D. melanogaster*; note that there is a large inversion from 84F to 93F in relation to *D. melanogaster*. Assay #2, significant regions are listed first for both a threshold of 0.05 and 0.1. n/a = this trait did not have a likelihood ratio that crossed the significance threshold. ^2^ The peak is the cytological location with the highest likelihood ratio (LR). ^3^ QTL effects are in phenotypic standard deviation units. ^4^ The *TR* is the proportion of the variance explained by that QTL when cofactors are taken into account.

## Data Availability

All data are included in the manuscript and Appendix A.

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
