# Peer review of "Intraspecific Genetic Variation for Behavioral Isolation Loci in Drosophila"

_genes, 2021, doi:10.3390/genes12111703_

Round 1

Reviewer 1 Report

This paper asks if the intraspecies variation in behavioral isolation is affected by the same genomic regions that result in interspecies behavioral isolation. The authors test how receptive D. mauritiana females are to D. simulans males that originate from different geographic regions. They repeat the QTL mapping twice, beginning with a subset of 3 strains and expanding to include 8 different strains, including repeating the initial 3 to look for replication.

I thought the experiment was well designed to express the question that was posed and I particularly liked that the authors replicated the QTL mapping of the same strains to see how repeatable the findings were. The authors state clear predictions for this study, although those predictions did not occur. They did not find overlap in regions across strains or repetition within the strain across experiments. However, they did a nice job presenting the results and discussing the possible biological interpretation and did not overstate the significance of their results.

Author Response

Thank you for this positive review.

Reviewer 2 Report

This is a very well executed replication study of previously identified QTLs (Moehring et al. 2004) that are essential for the separation of two closely related Drosophila species. The study not only tries to replicate those findings, but rather aims at exploring the replicability of findings across multiple populations. Although the results (that the QTL effects identified by Moehring et al. (2004) cannot be replicated) may seem daunting, and reminiscent of the wider replicability crisis, I find the current study highly valuable in clarifying what we know about species isolation. As the study is powerful and well-executed it appears very worth publishing.

Given that the study is well done and the paper is well written, most of my comments are fairly minor, but maybe I have one more important comment to make, which the Authors could address in the Discussion section, and possibly by presenting some additional analyses (if deemed feasible and useful).

One striking and somewhat puzzling result (see p-value in line 274) is that the highly significant QTL effects of the initial study from 2004 are very different from the effects observed here in assay 1 and assay 2. Line 276 refers to 3-way contingency tables underlying these tests, but I was not able to find all of the corresponding data in the Supplement. Supplementary Tables 1 and 2 are excellent for inspecting the results from assays 1 and 2 (e.g. using the pivot table function), but the data from the 2004 study does not seem to be available, which would be useful to have. Now, I was wondering, how such an analysis of contingency tables would turn out, if the data from 2004 was excluded from the analysis. Are the QTL signals from assay 1 significantly different from those of assay 2, or is it that signals tended to be similar but sometimes fell above and sometimes below the significance cutoff? Likewise, considering the data within one assay, e.g. the 8 strains from assay 2: are the QTL-effects significantly different between strains or not? This would be important to know for a better understanding of why QTL signals cannot be replicated. To interpret this “failure to replicate”, it might be important to know whether the copulation success of a given BC1 male is really a repeatable trait of that male (due to its distribution of 25% simulans DNA), or whether copulation success strongly depends on the individual female that the male is tested with. I am not sure whether this male repeatability is known to the Authors, but it might be important for explaining the difficulty of QTL mapping the responsible male trait loci. Finally, I noticed that mean copulation success in assay 1 was surprisingly low (31% compared to 68% in assay 2, and given that F1 males, with 50% simulans DNA, already achieved 36% copulation success in Table 1). What was the rate in the 2004 study? Is it possible that the varying responsiveness of females is partly responsible for the difficulty of mapping the male trait loci?

Minor comments:

  1. Introduction: For the ease of following, it could be made clear earlier on that QTL mapping can be carried out in the females or in the males (I think the 2004 study did both), and that here you limit yourself to mapping in males (assuming little variance in female rejection).
  2. Line 95: “a strain conferring a relatively high mating frequency for F1 males”. This is hard to understand at this point of the paper. Hence, you could either consider simplifying or omitting this information, or explain it by adding something like “(courting hybrid male getting accepted by pure mauritiana females)”.
  3. Line 148: explain “groupings of 20-30”. Is this one observer monitoring these simultaneously or consecutively in one day?
  4. Line 159: maybe clarify that latency is only scored if the event occurs (missing values otherwise, rather than attributing a minimum latency of the duration of observation).
  5. Line 222: P=0.1 not 0.01
  6. Line 235: It would be helpful to say early on that you consider epistasis between two loci in the same male (as one could think of loci in the female too).
  7. Line 254: “almost all”: specify the mean percentage or make a Supplementary Table showing each Line (similar to Table 1).
  8. Line 265: The presence/absence of food should be explained in the methods (maybe I just missed it). Was there a reason for changing the setup?
  9. Line 266: “plus five additional strains”. At this point, after showing Table 1, it might be easier to say “the remaining five strains in Table 1”
  10. Line 273: change to “using either the same (round 1) or different (round 2) assay conditions” (or the other way around, whichever is correct).
  11. Line 299-300: parentheses (plural needed?)
  12. Figure 1: Consider illustrating at the top of the figure the rough microsatellite marker positions for each of the three studies (2004, assay 1, assay 2).
  13. Line 382: I would cut the “indeed are unlikely to”, since this seems to be an exaggeration.
